# Quantitative magnetic resonance imaging indicates brain tissue alterations in patients after liver transplantation

**Lukas Laurids Goede**[1,2,3☉], **Henning Pflugrad**[1,3☉], **Birte Schmitz**[2], **Heinrich Lanfermann**[2], **Anita Blanka Tryc**[1,3], **Hannelore Barg-Hock**[4], **Jürgen Klempnauer**[4], **Karin Weissenborn**[1,3‡], **Xiao-Qi Ding**[2‡]*

1 Department of Neurology, Hannover Medical School, Hannover, Germany, 2 Institute of Diagnostic and Interventional Neuroradiology, Hannover Medical School, Hannover, Germany, 3 Integrated Research and Treatment Centre Transplantation (IFB-Tx), Hannover Medical School, Hannover, Germany, 4 Clinic for Visceral and Transplant Surgery, Hannover Medical School, Hannover, Germany

☉ These authors contributed equally to this work.
‡ These authors also contributed equally to this work.
* ding.xiaoqi@mh-hannover.de

**Data Availability Statement:** All relevant data are within the paper and its Supporting Information files.

## Abstract

### Purpose

To investigate cerebral microstructural alterations in patients treated with calcineurin inhibitors (CNI) after orthotopic liver transplantation (OLT) using quantitative magnetic resonance imaging (qMRI) and a cross-sectional study design.

### Methods

Cerebral qMRI was performed in 85 patients in a median 10 years after OLT compared to 31 healthy controls. Patients were treated with different dosages of CNI or with a CNI-free immunosuppression (CNI-free: n = 19; CNI-low: n = 36; CNI-standard: n = 30). T2-, T2*- and T2'- relaxation times, as well as apparent diffusion coefficient (ADC) and fractional anisotropy (FA) were measured in brain gray and white matter by using the regions of interest method.

### Results

In comparison to controls, patients revealed significantly increased T2, T2*, T2', ADC and reduced FA, predominantly in the frontal white matter, indicating microstructural brain alterations represented by increased free water (increased T2), reduced neuronal metabolism (increased T2') and a lower degree of spatial organization of the nervous fibers (reduced FA). CNI-low and CNI-free patients showed more alterations than CNI-standard patients. Analysis of their history revealed impairment of kidney function while under standard CNI dose suggesting that these patients may be more vulnerable to toxic CNI side-effects.

**Funding:** This study was supported by a grant from the German Federal Ministry of Education and Research (reference number: 01EO1302) to LLG and German Research Foundation (DFG) to BS. We acknowledge support by the German Research Foundation (DFG) and the Open Access Publication Fund of Hannover Medical School (MHH). The funders had no role in study design, data collection and analysis, decision to publish, or preparation of the manuscript.

**Competing interests:** The authors have declared that no competing interests exist.

**Abbreviations:** CNI, calcineurin inhibitors; FA, fractional anisotropy; fWM, frontal white matter; GCC, genu corpus callosum; GFR, glomerular filtration rate; MMF, mycophenolate mofetil; MS, multiple sclerosis; NAWM, normal appearing white matter; OLT, orthotopic liver transplantation; pWM, parietal white matter; qMRI, quantitative magnetic resonance imaging; RBANS, Repeatable Battery for the Assessment of Neuropsychological Status.

## Conclusion

Our findings suggest that the individual sensitivity to toxic side effects should be considered when choosing an appropriate immunosuppressive regimen in patients after liver transplantation.

## Introduction

Liver transplantation requires life-long immunosuppression. Calcineurin inhibitors (CNI), cyclosporine A and tacrolimus, are the most important agents in the immunosuppressive therapy regime besides mycophenolate mofetil (MMF), mTor inhibitor everolimus and steroids for the prevention of graft rejection in patients after orthotopic liver transplantation (OLT) [1]. The survival rates after OLT have increased significantly since the implementation of CNI into the therapy regimen [2]. Consequently, the long-term side effects of maintenance immunosuppressive therapy have received more attention lately. These may impact the quality of life and contribute to long-term morbidity and mortality of the patients. Numerous studies reported on renal dysfunction, malignancy and cardiovascular disease related to long-term CNI therapy, e.g. an increased risk of chronic kidney disease of up to 28%, a cardiovascular disease events rate of 24% and a cumulative cancer incidence of 13–26% at about ten years after transplantation as reviewed by Aberg et al.[3]. Furthermore, neurological complications including confusion, hallucinations, somnolence, stupor or seizures were observed in one third of the patients in the early course after OLT and were attributed to the neurotoxicity of CNI [4–7]. By using cerebral magnetic resonance imaging (MRI) increased ventricle volumes and progressive focal white matter lesions were observed in patients 6 to 9 years after OLT [8]. These findings indicate possible long-term side effects of CNI with impact on brain tissue in patients after OLT.

However, more sophisticated methods are needed to get further insight into the pathomechanism of CNI neurotoxicity. It has been reported that quantitative MR measurements are sensitive to certain pathological or physiological microstructural alterations that are usually invisible in conventional MRI [9–11]. With quantitative MRI (qMRI) measurements, such as MR relaxometry, the relaxation processes of the brain tissue can be quantified. For example, the transverse relaxation due to spin-spin interactions characterized by the irreversible (T2), the susceptibility effects of local magnetic field inhomogeneity characterized by the reversible relaxation time (T2') or both mechanisms together characterized by apparent (T2*) relaxation times, with the relation of $1/T2^* = 1/T2 + 1/T2'$, can be quantified. Moreover, the apparent diffusion coefficient (ADC) and the fractional anisotropy (FA), derived from diffusion tensor imaging (DTI), measure the proton diffusional activity within the tissue structures. The variations of these quantitative parameters reflect alterations in the molecular environment within the brain tissue. Thus, these parameters provide information about microstructural alterations that are often invisible in conventional MRI [9–13]. A first relaxometry measurement in patients after OLT has been reported by Herynek et al., who found increased T2 relaxation times in the thalamus and white matter in patients up to 15 years after OLT. The authors attributed this to damage caused by permanent exposure to immunosuppressants [14]. Therefore, we conducted this single center observational study to investigate brain functional, metabolic and microstructural alterations associated with long-term effects of CNI treatment in different doses in patients after OLT by using combined psychometric assessment and neuroimaging methods. Here we mainly report about the results obtained by using qMRI including

MR relaxometry and diffusion tensor imaging (DTI), while the detailed results of psychometric testing and morphological MRI have been reported elsewhere [15]. We hypothesized that microstructural brain tissue alterations measured by qMRI are associated with the administered dose of long-term CNI therapy in patients after OLT.

## Methods

The research protocol was approved by the ethics committee at Hannover Medical School. Informed consent was obtained from all patients and control subjects. All measurements were performed in accordance to the ethical guidelines of the World Medical Association Declaration of Helsinki (revised in 2008). None of the transplant donors were from a vulnerable population and all donors or next of kin provided written informed consent that was freely given.

### Subjects

The patients were recruited for the study as previously described in detail [15]. In summary, all patients treated in the liver transplantation outpatient clinic of Hannover Medical School were screened for suitability. All included subjects were enrolled between February 2014 and February 2016. Inclusion criteria for patients were an age of between eighteen and eighty years, a time interval of at least two years since liver transplantation and a stable immunosuppressive therapy of at least two years. Exclusion criteria were: an age of under eighteen years at the time of transplantation, an additional transplant of organs other than the liver, re-transplantation more than three months after primary transplantation, pre-existing neurological or psychiatric diseases that might affect brain structure or function, as for example a history of stroke, neurodegenerative diseases or depression, contraindications for MRI, a daily intake of prescribed drugs (besides CNI and steroids) such as antidepressants or antipsychotic medication which might affect brain function or microstructure of the brain, acute organ rejection or acute infection and decompensated heart-, liver- or kidney function.

To address potential sources of bias, the patients of the different groups were adjusted to age, sex, education and time since transplantation. According to the design of the study it was intended to include 30 patients per group. A total 91 patients were enrolled. After application of the inclusion and exclusion criteria 85 patients finally took part in the study. All patients had been treated with a standard dose of CNI (CNI-standard) for a median 5 years after OLT. Later, those patients who showed an impairment of kidney function as a side effect of CNI therapy either received a reduced dose of CNI (CNI-low) or other medication for immunosuppression (CNI-free). This alteration of the treatment regimen took place in a median 4 years after OLT in the CNI low dose group and in the CNI-free group, respectively. To investigate a long-term impact of CNI therapy with varied doses after OLT, the patients were divided into three groups according to their CNI medication at the time of assessment (for details see Pflugrad et al., 2018 [15]): patients with an immunosuppressive therapy regime without CNI (group 1, CNI-free, n = 19), patients with reduced dose CNI therapy (group 2, CNI-low = stable tacrolimus blood trough level <5 µg/l or ciclosporine A blood through level <50 µg/l, n = 36) and patients with standard dose CNI therapy (group 3, CNI-standard = stable tacrolimus blood trough level ≥5 µg/l or ciclosporine A blood trough level ≥50 µg/l, n = 30).

In addition, thirty-one—healthy controls, adjusted according to sex, age and education, served as reference group (group 4). In comparison to the study published by Pflugrad et al. [15], the present paper has one patient less in group 1 (CNI-free) and two subjects less in group 4 (controls) due to incomplete qMRI examinations and one patient more in group 2 (CNI-low), who was not considered in the analysis of the psychometric results due to missing

data. Thus, all patients except one from the CNI-low group considered for this analysis were part of the analysis described in [15].

### Clinical assessment

Each subject underwent a neurological examination. Further, age, sex, underlying liver disease, presence of arterial hypertension, diabetes mellitus, glomerular filtration rate (GFR), hypercholesterolemia, a history of hepatic encephalopathy, the grade of hepatic encephalopathy at OLT, a history of post-transplant encephalopathy (neurological complications after OLT including disorientation, confusion, hallucinations, cognitive dysfunction, and seizures due to metabolic changes), years since OLT and CNI dosages as well as CNI trough levels of each visit at the Transplant Outpatient Clinic of Hannover Medical School were documented. For neuropsychological testing, the Repeatable Battery for the Assessment of Neuropsychological Status (RBANS) was used [16–18].

### MRI examination

MR examinations of all subjects were performed at 3T (Verio; Siemens, Erlangen, Germany) by using a 12-channel phased array head coil. The data were acquired with two sequences as described by Eylers et al. [19]—a transverse T2 weighted turbo spin echo (TSE) sequence with three echoes (triple TE) (TR/TE = 6640/8.7/70/131 ms; 150˚ flip angle; 256 x 208 matrix; 1 x 1 x 3 mm$^3$ voxel size, acceleration factor 2), and a transverse T2* weighted gradient (GRE) echo sequence with triple TE (TR/TE = 1410/6.42/18.42/30.42 ms; 20˚ flip angle; 256 x 208 matrix; 1 x 1 x 3 mm$^3$ voxel size; acceleration factor 2), and an additional transversal single-shot spin-echo echo-planar imaging (EPI) sequence for DTI with 12 motion-probing gradients (b = 0, 1500 s/mm2).

### Data processing and analysis

T2 weighted images were reviewed for structural abnormalities by two senior neuroradiologists. As previously described [19], parameter maps of the relaxation times T2 and T2* were obtained on-the-fly by the MR console with an extended image reconstruction, provided by the manufacturer, with monoexponential fitting to the signal-intensity decay curves of the triple TE data acquired with TSE and with GRE, respectively. ADC and FA maps were obtained from the EPI data. All the parameter maps were transferred to a work station and used for region of interest (ROI) measurements. Numeric values of qMRI parameters were measured from corresponding parameter maps in seventeen brain regions of interest (ROIs) in each brain hemisphere (examples are shown in Fig 1): 11 ROIs in cerebrum—in the frontal white matter (fWM) between the middle frontal gyrus and the frontal horn of lateral ventricle, the putamen, the pallidum, the parietal white matter (pWM) at the middle posterior gyrus, the caudate nucleus, the thalamus and the occipital grey matter, the anterior (GCC) and the posterior portion of the corpus callosum near the central axis, the semioval center above the lateral ventricle, the subcortical motor area in the hand knob region; 2 ROIs in the brainstem, the ventral and dorsal part of the pons; Four ROIs in the cerebellum—in the cerebellar peduncle, the cerebellar posterior lobe, as well as the superior and inferior pole of the cerebellar posterior lobe. All ROIs were carefully drawn on a single section of corresponding parameter maps as a circle with an area of 15 mm$^2$ within each brain structure, according to anatomic landmarks to minimize partial volume effects. The ROI measurements were performed using ImageJ software [20]. The measured data were controlled by quality criteria, i.e. only those values with a signal to noise ratio (SNR = the mean value over the ROI divided by the standard deviation) > 5 were considered for further analysis. To eliminate deviations of the signal intensity due to

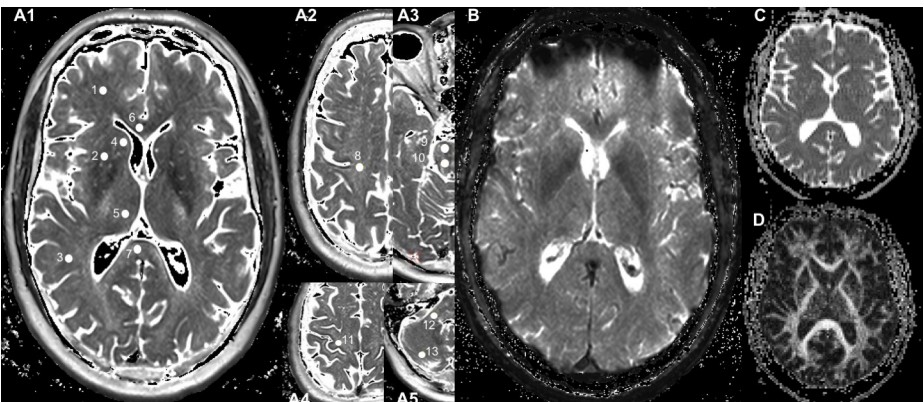

**Fig 1. Localization of regions of interest.** Quantitative parameter maps of the T2 map (A1-A5) the T2* map (B), the ADC map (C) and the FA map (D) are displayed. Filled circles (1–13) represent examples of ROI location measured in each hemisphere: frontal white matter (fWM, 1), putamen (2), parietal white matter (pWM, 3), caudate nucleus (4), thalamus (5), genu (GCC, 6) and splenium of corpus callosum (7), centrum semiovale (8), brainstem ventral (9) and dorsal (10), subcortical motor area in the hand knob region (11), cerebellar white matter (12) and cerebellar posterior lobe (13).

lateralization of brain function, the measured values from paired right and left ROIs were averaged. The values of the relaxation times T2 and T2* were measured in milliseconds (ms), and the T2' values were calculated according to the relationship $1/T2' = 1/T2*-1/T2$, thus, also with the unit ms. The parameter ADC was measured in the unit $mm^2s^{-1}$. The parameter FA was measured only in white matter ROIs and is without unit.

## Statistical analysis

The data were tested for normal distribution using the Shapiro-Wilk-test. The two-sided t-test was performed to compare the qMRI parameter values obtained from all patients (patient collective) to those of the controls. Corrections for multiple comparisons were performed by using the false-discovery rate (FDR) method with a desired false-discovery rate of 0.05.

One-way analysis of variance (one-way ANOVA) was used to compare the values of different groups. In cases with an overall $p < 0.05$, Dunnett's test at $\alpha = 0.05$ significance level was used for comparisons of each patient group with the control group. It is designed to hold the error rate at or below $\alpha$ when performing multiple comparisons of each treatment group with the control group [21]. Continuous abnormally distributed parameters were compared using the Kruskall-Wallis-Test and the Mann-Whitney U test. The Chi-squared test was used for testing categorical variables.

The results with a p-value below 0.05 were considered as statistically significant. Those with $0.05 < p < 0.08$ were considered as not significant but showing a tendency towards the tested alteration. All statistical analyses were performed using SPSS, Version 24 (IBM, Armonk, New York).

## Results

### Patient characteristics

The study groups were adjusted according to age, sex and education. Detailed characteristics of the patients and controls are given in Table 1, with minor modifications in comparison to the ones presented by Pflugrad et al. [15] because of slightly changed sample sizes as

**Table 1. Patient characteristics.**

| n = 116 | CNI free (group 1) n = 19 | CNI low (group 2) n = 36 | CNI standard (group 3) n = 30 | Control (group 4) n = 31 | p | p1vs3[1] |
|---|---|---|---|---|---|---|
| Age (years) mean ±SD | 60.74 (±9.1) | 59.67 (±9.4) | 54.8 (±10.1) | 58.48 (±7.8) | 0.093 | |
| Sex (male/female) | 14 (74%)/5 | 24 (67%)/12 | 18 (60%)/12 | 15 (48%)/16 | 0.277 | |
| Years after OLT median (IQ range) | 11 (3) | 10 (7) | 10 (9) | n. a. | 0.191 | |
| History of hepatic encephalopathy (+/-) | 4 (21%)15 | 4 (11%)/32 | 4 (13%)/26 | n. a. | 0.595 | |
| Grade of hepatic encephalopathy at OLT | 1 n = 3 3 n = 1 | 1 n = 2 2 n = 1 3 n = 1 | 1 n = 2 3 n = 2 | n.a. | 0.712 | |
| Post-transplant encephalopathy (+/-) | 2 (11%)/17 | 4 (11%)/32 | 7 (23%)/23 | n. a. | 0.314 | |
| Arterial hypertension (+/-) | 12 (63%)/7 | 23 (64%)/13 | 17 (57%)/13 | 11 (35%)/20 | 0.093 | |
| Diabetes mellitus (+/-) | 4 (21%)/15 | 8 (22%)/28 | 4 (13%)/26 | 1 (3%)/30 | 0.136 | |
| Hypercholesterolemia (+/-) | 7 (37%)/12 | 5 (14%)/31 | 7 (23%)/23 | 3 (10%)/28 | 0.083 | |
| GFR mean ±SD in ml/min | 63.21 (±19) | 74.36 (±23.96) | 84.27 (±24.23) | n. a. | **0.010** | **0.007** |
| Aetiology of liver disease (n) | | | | | | |
| Hepatitis B | 3 | 4 | 6 | n. a. | | |
| Hepatitis C | 1 | 0 | 0 | n. a. | | |
| Autoimmune (AIH, PBC, PSC) | 3 | 16 | 11 | n. a. | | |
| Alcohol | 2 | 1 | 1 | n. a. | | |
| Acute liver failure | 1 | 2 | 2 | n. a. | | |
| others | 9 | 13 | 10 | n. a. | | |

[1] Tukey post-hoc testing of group 1 vs group 3. n, number; CNI, calcineurin inhibitors; SD, standard deviation; OLT, orthotopic liver transplantation; IQ range, interquartile range; GFR, glomerular filtration rate; AIH, autoimmune hepatitis; PBC, Primary biliary cholangitis; PSC, primary sclerosing cholangitis

mentioned above. The comparison of GFR values showed that, at the time of recruitment, the patients in group 1 (CNI-free) had the most limited kidney function with the lowest mean GFR value, the patients in group 2 (CNI-low) had a less limited kidney function with a moderate GFR value, while the patients in group 3 (CNI-standard) had the best kidney function of all patient groups (p = 0.007). This observation is not surprising since the change of the immunosuppressive therapy from standard dose to low dose or CNI free therapy had been made predominantly because of increasing impairment of kidney function under standard dose CNI therapy [15].

In addition to CNI, mainly MMF and prednisolone served as immunosuppressive agents in the presented patients. The therapy regimen included MMF in sixteen patients from group 1 (CNI-free), in eighteen patients from group 2 (CNI-low) and in eight patients from group 3 (CNI-standard). Prednisolone was used in fourteen cases from group 1, in nine cases from group 2 and in ten cases from group 3. Three patients from group 1 took sirolimus, two patients took everolimus and one patient from group 2 took azathioprine.

Significant complications had occurred during OLT surgery or while on the intensive care unit in 44 (52%) of 85 patients. The patients with significant complications such as a need for additional surgery, transplant failure or infection were equally distributed between the three patient groups: Eighteen (60%) of the 30 patients of the CNI standard group (number of complications: one n = 12, two n = 3, three or more n = 3), sixteen (44%) of the 36 patients of the CNI low group (number of complications: one n = 8, two n = 6, three n = 2) and ten (53%) of the 19 patients of the CNI free group (number of complications: one n = 6, two n = 2, three n = 2) had had significant complications after OLT.

## Cognitive function

Although the subject sample sizes were slightly different, the RBANS results are similar to those previously described by Pflugrad et al.[15]: ANOVA showed significant group differences in the RBANS score of visuospatial and constructional ability (p = 0.005). Post hoc pairwise comparisons revealed significantly worse results in group 2 (CNI-low, 89 vs. 112, p = 0.007), and also in group 3 (CNI-standard, 96 vs 112, p = 0.026) compared to controls. The patients in group 1 (CNI free) also showed lower scores than the healthy controls but, however, the level of significance was missed. Moreover, the patients in group 2 (CNI-low) revealed significantly worse results than the controls (92 vs. 103, p = 0.008) in the RBANS total scale (including the results of all sub-categories: immediate memory, visuospatial and constructional ability, language, attention and delayed memory), while there was no significant difference between the other two patient groups and controls (S1 Table). The twelve patients (14%) with a history of HE before OLT did not differ from the patients without a history of HE (n = 73) in the RBANS with the exception of the RBANS subdomain language. Here patients with HE performed significantly worse than those without (91.8±14.8 vs 101.0±11.9, p = 0.02).

## Quantitative MRI measurements

**Comparison between the patient collective and the controls.** The Two-sided t-test revealed significant differences between the patient collective and the controls in the measured values T2, T2*, T2', ADC and FA in certain brain areas as shown in Table 2: In comparison to the controls higher values of relaxation times T2, T2* and T2' were found in patients in the fWM (p = 0.002 for T2, p = 0.001 for T2*, and p = 0.011 for T2'). Although higher values of T2, T2* and T2' were also observed in the patients compared to controls in the GCC and subcortical gray matter (pallidum, putamen, and thalamus) the level of significance was not reached (Table 2 and S2 Table). At the same time, higher ADC values in pWM (p = 0.001) were found in patients. Additionally, patients revealed lower FA values in fWM (p = 0.00005). Several patients' data were not considered for analysis due to minor quality according to the data quality criteria (SNR > 5). Missing data is indicated by the sample size displayed as N given in Table 2. The resulting smaller sub-groups of the patients kept the same characteristics

**Table 2. Significant results of two-sided t-tests for parameter values measured in patients and controls[1].**

| Brain[2] region | Patients | | | Controls | | | *p* |
|---|---|---|---|---|---|---|---|
| | N | Mean | SD | N | Mean | SD | |
| | | | T2 (ms) | | | | |
| fWM | 85 | 110.36 | 7.95 | 31 | 105.07 | 7.22 | *0.002* |
| | | | T2* (ms) | | | | |
| fWM | 85 | 44.26 | 4.90 | 31 | 40.83 | 4.91 | *0.001* |
| | | | T2' (ms) | | | | |
| fWM | 85 | 74.81 | 13.03 | 31 | 67.75 | 12.87 | *0.011* |
| | | | ADC ($\times 10^{-6}$ mm$^2$s$^{-1}$) | | | | |
| pWM | 85 | 709.16 | 46.01 | 31 | 678.45 | 37.99 | *0.001* |
| | | | FA | | | | |
| fWM | 69 | 0.351 | 0.056 | 31 | 0.406 | 0.066 | *0.000* |

[1] Corrections for multiple comparisons were performed by using the false-discovery rate (FDR) method, with the desired false-discovery rate to 0.05.

[2] Brain regions were the frontal WM (fWM) and the parietal white matter (pWM). Several patients' data were not considered for the parameter FA analysis due to minor quality according to data quality criteria (SNR > 5). n, number; SD, standard deviation

concerning sex, age and education. Overall, in comparison to the controls the patient collective showed significant regional brain alterations. Most changes occurred in the fWM (higher T2, T2*, T2' values and lower FA values and less in pWM (higher ADC values).

Patients with a history of HE before liver transplantation did not differ significantly from those without a history of HE concerning all measured qMRI data.

**Comparison of each patient group to controls.** The One-way ANOVA analysis revealed significant differences concerning qMRI parameter measurements across the patient groups in reference to the control group. The significant results of the ANOVA and post hoc Dunnett's tests are presented in Table 3, also including not significant results in brain regions where significant alterations were observed in the whole patient collective (Table 2): In comparison to the control group, patients in group 1 (CNI-free) and group 2 (CNI-low) showed increased T2 (p = 0.004 for group 1 and p = 0.030 for group 2) and T2* values (p = 0.018 for group 1 and p = 0.011 for group 2) in the fWM. In GCC, a tendency of increased T2 values in both group 1 (CNI-free, p = 0.055) and group 2 (CNI-low, p = 0.075), was observed, with a significantly increased T2* value (p = 0.020) being found only in CNI-low patients. No significant differences of T2' values were found between patient groups and controls, but a tendency of increased T2' values in fWM was observed in group 2 (p = 0.059 for overall test and 0.070 for post hoc test). Moreover, the patients in all three groups revealed a significantly lower value of FA in fWM (p = 0.001–0.020). Concerning ADC measurements, the patients in group 2 (CNI-low) showed an increased value in pWM (p = 0.001).

## Discussion

This observational study investigated 85 patients in a median 10 years after OLT under different CNI treatment regimens and 31 healthy controls by using qMRI measurements in multiple brain areas. We found significant differences between the patient collective and the controls as well as between different patient groups and the control group in several brain areas.

**Table 3. Significant results and results showing a tendency of tested alterations derived from ANOVA analysis on parameter values measured in different patient groups and the control group[1,2].**

| Brain[3] region | Para-meter | Group 1 (CNI-free) | | | Group 2 (CNI-low) | | | Group 3 (CNI-standard) | | | Group 4 (Controls) | | | ANOVA | Dunnett post-hoc testing[4] | | |
|---|---|---|---|---|---|---|---|---|---|---|---|---|---|---|---|---|---|
| | | N | Mean | SD | N | Mean | SD | N | Mean | SD | N | Mean | SD | p | p1vs4 | p2vs4 | p3vs4 |
| fWM | T2 | 19 | 112.8 | 10.4 | 36 | 110.4 | 7.5 | 30 | 108.7 | 6.4 | 31 | 105.1 | 7.2 | *0.004* | *0.004* | *0.030* | |
| GCC | T2 | 19 | 100.2 | 7.6 | 36 | 99.2 | 7.9 | 30 | 95.7 | 5.5 | 30 | 95.1 | 5.6 | *0.014* | *0.055* | *0.075* | |
| | | | | | | | | | | | | | | | | | |
| fWM | T2* | 19 | 45.1 | 5.5 | 36 | 44.6 | 5.0 | 30 | 43.3 | 4.4 | 31 | 40.8 | 4.9 | *0.006* | *0.018* | *0.011* | |
| GCC | T2* | 19 | 40.6 | 6.9 | 36 | 41.3 | 4.8 | 30 | 39.2 | 6.0 | 30 | 37.3 | 4.3 | *0.026* | | *0.020* | |
| | | | | | | | | | | | | | | | | | |
| fWM | T2' | 19 | 76.2 | 13.8 | 36 | 75.7 | 13.0 | 30 | 72.9 | 12.8 | 31 | 67.7 | 12.9 | *0.059* | | *0.070* | |
| | | | | | | | | | | | | | | | | | |
| pWM | ADC | 19 | 706.4 | 50.1 | 36 | 720.6 | 47.9 | 30 | 697.1 | 38.5 | 31 | 678.4 | 38.0 | *0.002* | | *0.001* | |
| | | | | | | | | | | | | | | | | | |
| fWM | FA | 16 | 0.351 | 0.042 | 30 | 0.347 | 0.056 | 23 | 0.357 | 0.066 | 31 | 0.406 | 0.067 | *0.001* | *0.019* | *0.001* | *0.020* |

[1] In cases of an overall p < 0.05, the Dunnett's test at α = 0.05 significance level was used for comparisons of each patient group with the control group. Cases with 0.05 p < 0.08 were considered as not significant but showing a tendency towards the tested alteration.

[2] The non-significant results in brain regions were also included where the significant alterations in the patient collective were observed (see Table 2).

[3] The selected brain regions are the frontal WM (fWM), the anterior part of the corpus callosum (GCC), and the parietal white matter (pWM).

[4] Results of the Dunnett's test of group 1 vs group 4 (*p1vs4*), group 2 vs group 4 (*p2vs4*) and group 3 vs group 4 (*p3vs4*). n, number; CNI, calcineurin inhibitors

The comparison between the patient collective and controls showed significantly increased brain T2, T2* and T2' values in patients in the frontal white matter. Also, higher, though not significant, values in the genu of corpus callosum and the subcortical gray matter (pallidum, putamen, and thalamus) were found. The observed T2 alterations in the patients are consistent with those reported by Herynek et al., who found increased T2 relaxation times in the white matter and in the thalamus in patients up to 15 years after OLT [14]. The parameter T2 is sensitive to microstructural variations in tissue like changes of the free water content, i.e. reduced free water corresponds to shortened T2 in the maturing brain of infants [22,23], while pathological demyelination or neurodegeneration resulting in increased free water content is associated with prolonged T2 [9,13,24]. Consequently, the observed increase of T2 in our patients may therefore reflect an increase of free water in the fWM. To our knowledge this study is the first to estimate brain reversible T2' values in patients after OLT. The parameter T2' reflects changes in the molecular level that influence brain local magnetic field homogeneity, e.g. those caused by a varied local concentration of deoxyhemoglobin [25,26]. An increase of T2' in frontal normal appearing white matter (NAWM) of patients with multiple sclerosis (MS) was explained by decreased oxygen extraction due to reduced metabolism. This assumption was consistent with observations in PET studies [27,28]. An increase of T2' in association with a decrease of neuronal metabolism was also observed in the splenium of the corpus callosum in a normal aging human brain [19]. Accordingly, the present observation of increased T2' in the fWM may indicate a reduction of metabolism in this brain area of our patients. Moreover, the alteration of T2' values seems to be associated with the brain function of the patients.

In addition, the patient collective revealed significantly higher ADC values in the parietal white matter and lower FA values in the frontal white matter compared to the controls. ADC indicates the movement scale of water molecule diffusion and the FA measures the degree of spatial organization of the nervous fiber structures [29]. Decreased ADC values were reported in tissues with high cellularity, e.g. tumors or cytotoxic edema [30,31], while relatively higher ADC values were seen in tissues with chronic tissue injury. For example, chronic hypoxia in patients with Eisenmenger syndrome led to higher ADC values in the frontal white matter and the lentiform nucleus compared to the healthy controls. This observation was explained by increased free water due to damages of myelin and axons in a state of chronic hypoxia [32]. Filippi et al. reported that the average lesion mean diffusivity (equivalent to ADC) was higher and the average lesion FA was lower than in the corresponding quantities of NAWM in patients with MS, while in NAWM of these patients the mean diffusivity (equivalent to ADC) was higher and the average lesion FA was lower compared to healthy controls. The findings were interpreted as severe tissue damage in MS lesions and microstructural changes in the NAWM of MS patients [33]. In line with these observations the present findings of higher ADC values in the parietal white matter of the patients may also be attributed to increased free water and the lower FA values in frontal white matter may be attributed to altered spatial organization of the nervous fiber structures.

Comparing the results given in Tables 2 and 3, it is clear that significantly reduced FA values in the fWM were not only present in the whole patient collective (Table 2) but also in each of the three patient groups (Table 3). In contrast, the significant increase of T2' that was observed in the whole patient group had no correspondence in the different patient subgroups. This probably indicates that these alterations were weaker than the FA alterations in the fWM. The further observed qMRI alterations in the patient collective were mainly found in group 2 (CNI-low), i.e. increased T2* values in the GCC and increased ADC values in the pWM, or in group 2 (CNI-low) and group 1 (CNI-free) together, i.e. increased T2 values in the fWM and the GCC, increased T2* values in the fWM. In comparison to the control group the patient groups receiving different doses of CNI revealed varied grades of deviations in the measured

parameters in addition to those found in all groups (lower FA values in frontal white matter). These observations indicate that, while patients in all three patient groups revealed altered spatial organization of the nervous fiber structures in the fWM (indicated by decreased FA values), the patients treated with CNI low dose showed the most significant brain microstructural variations (in fWM, GCC, and pWM). The patients without CNI therapy revealed additional significant brain microstructural variations (in fWM) but, however, less than the CNI low group. Patients treated with CNI standard dose did not have additional brain microstructural alterations. The observation that the patients treated with CNI low dose show the most extensive microstructural alterations is consistent with the psychometric results which showed that only the patients in group 2 (CNI-low) revealed significantly worse results regarding RBANS total scale (including the sub-categories immediate memory, visuospatial and constructional ability, language, attention and delayed memory) than the healthy controls.

First it was somewhat surprising that not the patients receiving a standard dose of CNI but those receiving a lower dose of CNI revealed the most microstructural alterations in the brain and showed cognitive impairment compared to the controls. However, considering that all patients were initially treated with a standard dose of CNI in a median of four years before the applied dose of CNI was reduced (CNI-low) or stopped (CNI-free), these observations might be interpreted as follows: The patients receiving OLT may react differently to the toxic side effects accompanied by CNI therapy. Some of the patients, especially those of patient group 3, who were continually treated with the CNI standard dose since transplantation, seemed to be quite resistant to the toxic side effects. Some of the patients, especially those of patient groups 1 (CNI-free) and 2 (CNI-low), however, appeared to be vulnerable to the toxic side effects of CNI. The initially applied standard CNI dose induced not only the initially observed kidney damage, but possibly also alterations of brain microstructures, which seem to be long-lasting. However, if CNI therapy was stopped, these brain alterations might recover partly as indicated by our patients with CNI free therapy who revealed less additional brain alterations than the patients treated with a low dose of CNI. Despite the fact that at the moment the determinative factors behind the different behavior of toxic side effects of CNI in patients remains unclear, our observations suggest to be cautious treating those patients who show any CNI side effects during therapy.

The CNI standard group and the CNI low dose and CNI free group differed significantly with regard to their GFR at the time of the study. Thus, it is necessary to discuss possible effects of impaired kidney function upon the MRI results, since alterations of cognitive function as well as brain metabolism have been described for patients with chronic kidney disease. However, in this study, all patient groups had a mean GFR above 60ml/min which underlines that the patients had, if at all, only slight kidney impairment. Furthermore, patients receiving the CNI free immunosuppressive therapy had the most impaired kidney function at the time of CNI dosage change and at study inclusion, while patients in group 2 (CNI-low), who had a higher GFR that did not significantly differ from that of group 3 (standard dose), showed most of the significant tissue changes. Thus, it can be assumed that the impact of kidney function on our measurements is less critical than that of the assumed vulnerability towards CNI toxicity.

The results of our study are limited by the fact that other immunosuppressants, which were consistently taken by the patients in this study, may have had an impact on the qMRI results of the cerebral tissue. Furthermore, the standard immunosuppressive therapy regime after OLT includes CNI and therefore no group of patients with CNI-free immunosuppressive therapy from the first day after transplantation was available. Other limitations are that this was a single-center observational study with limited transferability to other centers and unfortunately no data from before OLT was available from our patients. Finally, the sample size limits the statistical power and a statement on clinical relevance.

In conclusion, this study showed altered spatial organization of the nervous fiber structures and a trend of reduced metabolism in the frontal white matter in patients after OLT, regardless of the used immunosuppressive therapy regimen. Patients who had developed severely impaired kidney function under initial treatment with standard dose CNI revealed additional brain microstructural alterations, especially if the CNI therapy was continued using a lower dose of CNI and not replaced.

## Supporting information

**S1 Table. Results of the cognitive function testing with the RBANS.**
(DOCX)

**S2 Table. Results of two-sided t-tests for parameter values measured in patients and controls.**
(DOCX)

## Acknowledgments

The authors would like to thank Andreas Manthey for language editing.

## Author Contributions

**Conceptualization:** Lukas Laurids Goede, Henning Pflugrad, Karin Weissenborn, Xiao-Qi Ding.

**Data curation:** Lukas Laurids Goede, Henning Pflugrad, Birte Schmitz, Anita Blanka Tryc, Hannelore Barg-Hock, Karin Weissenborn, Xiao-Qi Ding.

**Formal analysis:** Lukas Laurids Goede.

**Funding acquisition:** Karin Weissenborn, Xiao-Qi Ding.

**Investigation:** Lukas Laurids Goede, Henning Pflugrad, Karin Weissenborn, Xiao-Qi Ding.

**Methodology:** Henning Pflugrad, Karin Weissenborn, Xiao-Qi Ding.

**Project administration:** Karin Weissenborn.

**Resources:** Heinrich Lanfermann, Hannelore Barg-Hock, Jürgen Klempnauer, Karin Weissenborn, Xiao-Qi Ding.

**Software:** Birte Schmitz, Xiao-Qi Ding.

**Supervision:** Karin Weissenborn, Xiao-Qi Ding.

**Validation:** Xiao-Qi Ding.

**Visualization:** Lukas Laurids Goede.

**Writing – original draft:** Lukas Laurids Goede.

**Writing – review & editing:** Lukas Laurids Goede, Henning Pflugrad, Karin Weissenborn, Xiao-Qi Ding.

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
