## [Decision Letter · Decision Letter 0]

6 Aug 2019

PONE-D-19-14760

Quantitative magnetic resonance imaging indicates brain tissue alterations in patients after liver transplantation

PLOS ONE

Dear Dr. Ding,

Thank you for submitting your manuscript to PLOS ONE. After careful consideration, we feel that it has merit but does not fully meet PLOS ONE’s publication criteria as it currently stands. Therefore, we invite you to submit a revised version of the manuscript that addresses the points raised during the review process.

In addition to addressing the comments raised by the Reviewers, please also address/respond to the following points:

1. Please review the STROBE statement and ensure that all points have been addressed (https://www.strobe-statement.org/index.php?id=strobe-home). As of now, there are a number of aspects that are missing (e.g. reporting of time frame of enrolled participants, addressing potential sources of bias).

2. Please avoid the use of "Results not shown" (See the following for link for additional details: https://journals.plos.org/plosone/s/data-availability)

3. The word ‘gender’ is used throughout the manuscript (in addition to 'sex'). Usually, “sex” (the biological designation) is meant. “Gender” is the social construct and is rarely relevant in neurologic disease. Please revise the text to use “sex” rather than “gender” throughout.

4. Were there any changes to the MRI scanner (e.g. hardware, software) throughout the course of the study? If so, were patients and controls equally distributed before/after updates?

We would appreciate receiving your revised manuscript by Sep 14 2019 11:59PM. To enhance the reproducibility of your results, we recommend that if applicable you deposit your laboratory protocols in protocols.io, where a protocol can be assigned its own identifier (DOI) such that it can be cited independently in the future. For instructions see: http://journals.plos.org/plosone/s/submission-guidelines#loc-laboratory-protocols

We look forward to receiving your revised manuscript.

Kind regards,

Niels Bergsland

Academic Editor

PLOS ONE

Journal Requirements:

2. Thank you for submitting the above manuscript to PLOS ONE. During our internal evaluation of the manuscript, we found some text overlap between your submission and the following previously published works, on which you are an author: Eylers, Vanessa V., et al. "Detection of normal aging effects on human brain metabolite concentrations and microstructure with whole-brain MR spectroscopic imaging and quantitative MR imaging." American Journal of Neuroradiology 37.3 (2016): 447-454. The re-use of text from a previous publication may have implications for the copyright or license that applies to the published article. To avoid any concerns in this respect, we would recommend that you revise the manuscript, particularly outside the methods section, to either remove any text overlap where possible/appropriate, or to clearly indicate that the text reproduces information already reported in detail elsewhere (e.g. “As described in detail previously [ref],…”) citing the relevant sources.

4. We noted in your submission details that a portion of your manuscript may have been presented or published elsewhere: "Neither the submitted material nor portions thereof have been published previously or are under consideration for publication elsewhere.

Preliminary results of this study have been presented at the International Society for Hepatic Encephalopathy and Nitrogen Metabolism (ISHEN) 2017 in New Delhi, India, as a poster. The according abstract has been published in the Journal of Clinical and Experimental Hepatology in February 2017."

Reviewers' comments:

Reviewer's Responses to Questions

**Comments to the Author**

1. Is the manuscript technically sound, and do the data support the conclusions?

Reviewer #1: Yes

Reviewer #2: Yes

2. Has the statistical analysis been performed appropriately and rigorously? 

Reviewer #1: Yes

Reviewer #2: Yes

3. Have the authors made all data underlying the findings in their manuscript fully available?

Reviewer #1: Yes

Reviewer #2: Yes

4. Is the manuscript presented in an intelligible fashion and written in standard English?

Reviewer #1: Yes

Reviewer #2: Yes

5. Review Comments to the Author

Reviewer #1: Dr Goede et al. present a study using quantitative MRI for neurological assessment after liver transplantation (LT). They compare, more than 2 years after LT patients under normal dosage of CNI, patients under lowered dosage of CNI and patients without CNI.

Here are some comments:

- page 12, how were "pre-existing neurological or psychiatric diseases" defined ?

- page 12, how was "regular intake of drugs (besides CNI and steroids) which might affect brain function" defined?

- were possible per-operative or ICU complications taken in account ?

- the cause of reducing the dosage of CNIs could constitute a bias. This is however discussed in discussion

- page 13, maybe a comparison to patients that underwent renal grafting and being under CNIs would have be valuable

- page 15, there is a figure legend included in the manuscript but there is no figure

- it seems that the present study constitute an ancillary study of the study previously described in ref 15. Please, if it is so, clearly explain that the patients are the same that those described in a previous study

- page 18, were RBANS profiles different in patients that had previous to LT bouts of hepatic encephalopathy ?

- table 2 is difficult to understand since only significant results are presented

- "p14", "p24" "p34" are difficult to understand. Please provide are clearer way to describe this comparution: "p1v4" maybe ?

- the figure 1 is not clear. Please add in the A part, a A1 and so on in order to be more precise

-

Reviewer #2: This is a very important manuscript that describes concerning long term disorders that are secondary to immunosuppression on patients after liver transplant. Well designed study and methods are sound and clear for the results.

The only aspect that I see that is needing is the degree of liver decompensation prior to liver transplant. The other factor that I think is missing is the effect of decompensated liver disease in the brain and different areas of the central nervous system. If there is predisposition for this deleterious effect of the CNI in a previously injured CNS.

6. PLOS authors have the option to publish the peer review history of their article (what does this mean?). If published, this will include your full peer review and any attached files.

Reviewer #1: Yes: Nicolas WEISS, MD, PhD

Reviewer #2: No

---

## [Author Response · Author response to Decision Letter 0]

19 Aug 2019

Dear Professor Bergsland, dear Reviewers,

Thank you very much for the critical review of our manuscript. In the following we would like to respond to each of the comments. 

Academic editor:

1. Please review the STROBE statement and ensure that all points have been addressed (https://www.strobe-statement.org/index.php?id=strobe-home). As of now, there are a number of aspects that are missing (e.g. reporting of time frame of enrolled participants, addressing potential sources of bias).

Answer: We reviewed our manuscript according to the STROBE statement for observational studies. The missing aspects were added to the manuscript. Please excuse that we do not list all changes here as they are scattered throughout the manuscript. 

2. Please avoid the use of "Results not shown" (See the following for link for additional details: https://journals.plos.org/plosone/s/data-availability)

Answer: Thank you for the advice. The missing data is now shown in supplemental table 2. 

3. The word ‘gender’ is used throughout the manuscript (in addition to 'sex'). Usually, “sex” (the biological designation) is meant. “Gender” is the social construct and is rarely relevant in neurologic disease. Please revise the text to use “sex” rather than “gender” throughout.

Answer: Thank you for the advice. The word ‘gender’ was changed to ‘sex’ throughout the manuscript. 

4. Were there any changes to the MRI scanner (e.g. hardware, software) throughout the course of the study? If so, were patients and controls equally distributed before/after updates?

Answer: During the course of the study no changes concerning hardware or software were made to the MRI scanner. All subjects underwent the same study protocol. 

Journal Requirements:

Answer: We changed our manuscript accordingly.

2. Thank you for submitting the above manuscript to PLOS ONE. During our internal evaluation of the manuscript, we found some text overlap between your submission and the following previously published works, on which you are an author: Eylers, Vanessa V., et al. "Detection of normal aging effects on human brain metabolite concentrations and microstructure with whole-brain MR spectroscopic imaging and quantitative MR imaging." American Journal of Neuroradiology 37.3 (2016): 447-454. The re-use of text from a previous publication may have implications for the copyright or license that applies to the published article. To avoid any concerns in this respect, we would recommend that you revise the manuscript, particularly outside the methods section, to either remove any text overlap where possible/appropriate, or to clearly indicate that the text reproduces information already reported in detail elsewhere (e.g. “As described in detail previously [ref],…”) citing the relevant sources.

Answer: You are right. Parts of the MR examination and data processing of our study are similar to the MR processing described in the manuscript "Detection of normal aging effects on human brain metabolite concentrations and microstructure with whole-brain MR spectroscopic imaging and quantitative MR imaging”. We have added this information to the methods section of the manuscript and have included a reference (page 10).

3. We suggest you thoroughly copyedit your manuscript for language usage, spelling, and grammar.

Answer: The manuscript was reviewed to improve the language use, spelling and grammar with the help of Andreas Manthey who has a university degree in English language studies. 

4. We noted in your submission details that a portion of your manuscript may have been presented or published elsewhere: "Neither the submitted material nor portions thereof have been published previously or are under consideration for publication elsewhere.

Preliminary results of this study have been presented at the International Society for Hepatic Encephalopathy and Nitrogen Metabolism (ISHEN) 2017 in New Delhi, India, as a poster. The according abstract has been published in the Journal of Clinical and Experimental Hepatology in February 2017."

Answer: Preliminary results of this study were presented as a poster at the 17th International Society for Hepatic Encephalopathy and Nitrogen Metabolism (ISHEN) 2017 in New Delhi, India. Only the according abstract was published: 

Goede L. et al., Neurotoxic Side Effects of Calcineurin Inhibitors in Patients After Liver Transplantation: Preliminary Results of a Quantitative MRI Study of the Brain, Journal of Clinical and Experimental Hepatology, Volume 7, Supplement 1, February 2017, Pages S31-S32. 

The abstract was not peer-reviewed. This work does not constitute a dual publication and should be included in the current manuscript. 

Comments to the Author

Reviewer #1: Dr Goede et al. present a study using quantitative MRI for neurological assessment after liver transplantation (LT). They compare, more than 2 years after LT patients under normal dosage of CNI, patients under lowered dosage of CNI and patients without CNI.

Here are some comments:

- page 12, how were "pre-existing neurological or psychiatric diseases" defined ?

Answer: Pre-existing neurological or psychiatric diseases are defined as any neurological or psychiatric disease that might affect brain structure or function. At the time of inclusion in our study all patients were asked for diagnosed diseases and daily intake of medication that indicated a neurological or psychiatric disease. Furthermore, the records of the patients were screened for neurological or psychiatric diseases. If a neurological or psychiatric disease was found the patient was excluded from the study. To clarify this important point we have added a sentence to the methods section of the manuscript (page 6):

“…that might affect brain structure or function, as for example a history of stroke, neurodegenerative diseases or depression, …” 

- page 12, how was "regular intake of drugs (besides CNI and steroids) which might affect brain function" defined?

Answer: Drugs like antidepressants or antipsychotic medication were considered to affect brain function. All patients were asked which medication they take on a daily and the records of the patients were checked. If a drug that might affect brain function was identified the patient was excluded from the study. To clarify this exclusion criterion we have added a sentence to the methods section of the manuscript (page 6):

“…a daily intake of prescribed drugs (besides CNI and steroids) such as antidepressants or antipsychotic medication which might affect brain function or microstructure of the brain, …”

- were possible per-operative or ICU complications taken in account ?

Answer: Yes indeed. Of our 85 included patients 44 (52%) had significant complications during surgery or while on ICU such as sepsis or a need for further surgery. The distribution of patients with significant complications after liver transplantation did not differ between the patient groups. We added the respective data to the results section of the manuscript (page 12):

“Significant complications had occurred during OLT surgery or while on the intensive care unit in 44 (52%) of 85 patients. The patients with significant complications such as a need for additional surgery, transplant failure or infection were equally distributed between the three patient groups: Eighteen (60%) of the 30 patients of the CNI standard group (number of complications: one n=12, two n=3, three or more n=3), sixteen (44%) of the 36 patients of the CNI low group (number of complications: one n=8, two n=6, three n=2) and ten (53%) of the 19 patients of the CNI free group (number of complications: one n=6, two n=2, three n=2) had had significant complications after OLT.”

- the cause of reducing the dosage of CNIs could constitute a bias. This is however discussed in discussion

- page 13, maybe a comparison to patients that underwent renal grafting and being under CNIs would have be valuable

Answer: We agree that this is an interesting aspect which, however, should be addressed in the future because, unfortunately, we did not include patients that had kidney transplantation in our study. 

- page 15, there is a figure legend included in the manuscript but there is no figure

Answer: Please excuse the ambiguity. The legend of figure 1 was included after the paragraph of its first mention as demanded by the style requirements of PLOS ONE. The figure is displayed separately and not directly in the text. 

- it seems that the present study constitute an ancillary study of the study previously described in ref 15. Please, if it is so, clearly explain that the patients are the same that those described in a previous study

Answer: You are correct. The patients described in this study took part in a bigger study analyzing brain functional, metabolic and microstructural alterations associated with long-term effects of treatment with different doses of CNI. To clarify this we have added a sentence to the methods section of the manuscript (page 7):

“Thus, all patients except one from the CNI-low group considered for this analysis were part of the analysis described in [15].”

- page 18, were RBANS profiles different in patients that had previous to LT bouts of hepatic encephalopathy ?

Answer: Twelve of the 85 patients (14%) had a history of HE before OLT. Patients with an HE history did not differ from those without in the RBANS except in the RBANS subdomain language. This result was added to the manuscript in the results section (page 13):

“The twelve patients (14%) with a history of HE before OLT did not differ from the patients without a history of HE (n=73) in the RBANS with the exception of the RBANS subdomain language. Here patients with HE performed significantly worse than those without (91.8±14.8 vs 101.0±11.9, p=0.02).”

This result is interesting because impairment of language is typically not a feature of hepatic encephalopathy. However, this result should be interpreted with caution as the study was not designed to compare patients based on their history of hepatic encephalopathy.

- table 2 is difficult to understand since only significant results are presented

Answer: Thank you for this advice. We have added a supplemental table 2 with all results of the two-sided t-tests for parameter values measured in patients and in controls. For table 2 we extracted significant results using p<0.05 and false-discovery rate method for correction of multiple comparison. 

- "p14", "p24" "p34" are difficult to understand. Please provide are clearer way to describe this comparution: "p1v4" maybe ?

Answer: Thank you for this suggestion. We have added a “vs” between the numbers to improve the clarity. 

- the figure 1 is not clear. Please add in the A part, a A1 and so on in order to be more precise

Answer: We have adjusted figure 1 accordingly. 

Reviewer #2: This is a very important manuscript that describes concerning long term disorders that are secondary to immunosuppression on patients after liver transplant. Well designed study and methods are sound and clear for the results.

The only aspect that I see that is needing is the degree of liver decompensation prior to liver transplant. 

Answer: Thank you for addressing this important point. Twelve of the 85 patients had an acute episode of hepatic encephalopathy (HE) with grades between 1 to 3 at the time of liver transplantation, who were distributed equally between the patient groups each containing 4 of them: CNI standard group (HE grade 1, n=2; and grade 3 n=2), CNI low-dose group (HE grade 1, n=2; grade 2, n=1, and grade 3, n=1), CNI free group (HE grade 1, n=3; and grade 3, n=1). We have added this data to table 1. 

The other factor that I think is missing is the effect of decompensated liver disease in the brain and different areas of the central nervous system. If there is predisposition for this deleterious effect of the CNI in a previously injured CNS.

Answer: Thank you for addressing this important issue. We did not find significant differences of the qMRI measurements between the patients with and without a history of hepatic encephalopathy before liver transplantation. We have added this information to the results section of the manuscript (page 14):

“Patients with a history of HE before liver transplantation did not differ significantly from those without a history of HE concerning all measured qMRI data.”

Furthermore, to exclude an effect of structural brain alteration on our qMRI measurements all MRI images were checked for macroscopic brain tissue damage. If those were found, the patients were excluded from the study (page 8).

“T2 weighted images were reviewed for structural abnormalities by two senior neuroradiologists”

We would like to express our thanks for the review of our manuscript. We hope that we have satisfactorily addressed all comments made by the reviewers and the academic editor and that you will now find our manuscript acceptable for publication in PLOS ONE.

---

## [Decision Letter · Decision Letter 1]

11 Sep 2019

[EXSCINDED]

Quantitative magnetic resonance imaging indicates brain tissue alterations in patients after liver transplantation

PONE-D-19-14760R1

Dear Dr. Ding,

We are pleased to inform you that your manuscript has been judged scientifically suitable for publication and will be formally accepted for publication once it complies with all outstanding technical requirements.

With kind regards,

Niels Bergsland

Academic Editor

PLOS ONE

Additional Editor Comments (optional):

Reviewers' comments:

Reviewer's Responses to Questions

**Comments to the Author**

1. If the authors have adequately addressed your comments raised in a previous round of review and you feel that this manuscript is now acceptable for publication, you may indicate that here to bypass the “Comments to the Author” section, enter your conflict of interest statement in the “Confidential to Editor” section, and submit your "Accept" recommendation.

Reviewer #1: All comments have been addressed

2. Is the manuscript technically sound, and do the data support the conclusions?

Reviewer #1: Yes

3. Has the statistical analysis been performed appropriately and rigorously? 

Reviewer #1: Yes

4. Have the authors made all data underlying the findings in their manuscript fully available?

Reviewer #1: Yes

5. Is the manuscript presented in an intelligible fashion and written in standard English?

Reviewer #1: Yes

6. Review Comments to the Author

Reviewer #1: Thank to all the authors of the article. All my comments have been addressed correctly. I have no further remarks.

Nice work !

7. PLOS authors have the option to publish the peer review history of their article (what does this mean?). If published, this will include your full peer review and any attached files.

Reviewer #1: Yes: Nicolas WEISS

---

## [Editor Report · Acceptance letter]

18 Sep 2019

PONE-D-19-14760R1 

Quantitative magnetic resonance imaging indicates brain tissue alterations in patients after liver transplantation 

Dear Dr. Ding:

I am pleased to inform you that your manuscript has been deemed suitable for publication in PLOS ONE. Congratulations! Your manuscript is now with our production department. 

With kind regards,

on behalf of

Dr. Niels Bergsland 

Academic Editor

PLOS ONE